# Cytomegalovirus Retinitis in HIV and Non-HIV Individuals

**DOI:** 10.3390/microorganisms8010055

**Published:** 2019-12-28

**Authors:** Monique Munro, Tejabhiram Yadavalli, Cheryl Fonteh, Safa Arfeen, Ann-Marie Lobo-Chan

**Affiliations:** 1Department of Ophthalmology and Visual Sciences, University of Illinois at Chicago, Chicago, IL 60612, USA; 2Department of Microbiology and Immunology, University of Illinois at Chicago, Chicago, IL 60612, USA

**Keywords:** cytomegalovirus retinitis, HIV, organ transplant, immunosuppression

## Abstract

Cytomegalovirus retinitis (CMVR) is a severe, vision-threatening disease that primarily affects immunosuppressed patients. CMVR is the most common ocular opportunistic infection in human immunodeficiency virus (HIV) infected patients and is the leading cause of blindness in this group; however, the incidence of CMVR in HIV patients has dramatically decreased with antiretroviral therapy. Other causes of immunosuppression, including organ transplantation, hematologic malignancies, and iatrogenic immunosuppression, can also lead to the development of CMVR. Herein, we describe the pathogenesis of CMVR and compare clinical features, epidemiology, and risk factors in HIV and non-HIV infected individuals with CMVR.

## 1. Introduction

Cytomegalovirus (CMV) is a member of the Herpesviridae family [1,2]. CMV presents as an opportunistic infection in 75–85% of cases; and although infection can occur in healthy individuals, it is uncommon to observe symptomatic CMV infection in individuals who are not immune suppressed [3,4]. One of the most deleterious manifestations of CMV infection in the eye is CMV retinitis (CMVR). CMVR is characterized by full thickness retinal inflammation, hemorrhage, and necrosis [2]. CMVR can result in profound vision loss and may lead to retinal detachment and permanent vision loss. In the absence of known immune suppression, ocular manifestations of CMVR should raise suspicion for human immunodeficiency virus (HIV) infection and is classified as an acquired immunodeficiency syndrome (AIDS) defining illness [4,5]. CMVR may also be seen in other conditions associated with immunosuppression, including solid organ transplants, hematologic malignancy, and drug-induced immunosuppression [4,6].

In this narrative review, the pathogenesis, ocular manifestations, treatment and complications of CMVR in HIV and non-HIV infected individuals are presented. 

## 2. Pathogenesis of CMV

There is an estimated global CMV seroprevalence of 83% in the general population [7]. CMV infection may occur via multiple routes. Possible transmission paths include prenatal intrauterine infection, perinatal infection through breast milk or genital secretions, and postnatal exchange of bodily fluids such as blood, saliva, and sperm [8,9,10]. It has also been observed that CMV infection can occur during organ transplantation of bone marrow, heart, and lung [11,12].

Although CMVR in humans has been extensively studied through clinical studies, the lack of an animal model due to the species specificity of CMV hindered in vivo research; especially those that mimic HIV induced CMVR, for a long period of time [13]. The use of murine CMV (MCMV) to study ocular pathogenesis has been validated since the early 1980s in both C57BL6 and BALB/c mice [14]. By injecting the virus into the anterior chamber of the eye, scientists were able to study acute viral replication, persistence and the establishment of latency in various ocular tissues [14]. In order to study the pathogenesis in immunocompromised mice, irradiated/anti-lymphocyte serum treated BALB/c mice could be infected with MCMV and viral spread could be evaluated in comparison to immuno-competent mice [15]. While these models of immunosuppression revealed a good deal of insight into the pathogenesis of CMV in immunocompromised patients, none could reproduce the intricacy of HIV induced immune deficiency [13]. Using a retrovirus infection, a murine model of AIDS (MAIDS) characterized by progressive immune dysfunction, could be induced in small animal models. MAIDS induced mice when infected with CMV revealed up to 90% development of necrotizing retinitis within 10 days of infection. Interestingly, the pathogenesis of CMVR in MAIDS mice was similar to those observed in human counterparts [13,16].

Through sub-retinally injected MCMV in MAIDS mice, one could not only observe the progression of CMVR disease but could analyze for the presence and absence thereof for various cytokines, immune cell infiltration, immune-activation and antigen presentation, and mode of cell death during active and latent infection. One of the most important findings from these studies includes the upregulation of suppressor of cytokine signaling (SOCS) proteins SOCS1 and SOCS3 [16,17]. SOCS stimulation triggers a negative feedback in cell signaling via the regulation of Janus kinase (JAK)/signal transducer and activator of transcription (STAT) pathways initiated by antiviral interferons (IFN) and other cytokines such as interleukin (IL)-6 [16,17]. Viral activation of these host proteins help dysregulate host antiviral strategies and thereby assist virally-infected cells in evading immune destruction. Macrophages during MCMV infection, depending on their differentiation status or reaction to type I and type II interferons, play supportive or contradictory roles. It has been demonstrated that macrophages infected with MCMV become resistant to IFN-γ-induced activation due to the intracellular stimulation of SOCS family members, leading to efficient immune-evasion by the virus. Recent studies to uncover the cell death pathways have revealed that in addition to apoptosis, other pathways such as necroptosis and pyroptosis are simultaneously stimulated during MAIDS induced CMVR [18]. Although these studies shed new light on various underlying pathways through which CMV causes sight-threatening disease, many pathogenic causes are unresolved and remain to be uncovered through continued research.

### 2.1. Adult Transmission

In the general adult population, the transmission of CMV through bodily fluids is uncommon given the fact that the number of virus-shedding seropositive individuals is very low [19]. Like many other herpesviruses, once infected, CMV remains latent in peripheral blood leukocytes and bone marrow cells for the remainder of the individual’s life [20]. Bearing on the immune status of the individual, reactivation and the non-symptomatic shedding of the virus is almost negligible in individuals above the age of 30 [21]. In older populations of immune-suppressed and HIV infected patients, CMV reactivates and sheds from bodily fluids mentioned above [21,22]. 

In immunocompetent individuals, CMV remains asymptomatic but has the potential to reactivate when the immune system is compromised [19]. In immunocompromised patients, primary CMV infection causes severe complications such as pyrexia, viremic-septicemia, pneumonitis and in rare cases CMV induced immunosuppression [23,24]. CMV may also disseminate to the gastrointestinal tract and cause ulcerations. Many studies have hypothesized the hematogenous spread of CMV to the eye during immunosuppressed state, which disseminates to the retina resulting in retinitis [25].

### 2.2. Congenital Transmission

While only 1% of newborns in the United States are infected with CMV, up to 5%–10% of these patients demonstrate signs and symptoms at birth and 1%–2% suffer from severe complications that may lead to death [26,27,28]. Transmission may occur during CMV viremia in a pregnant woman allowing exposure of the placenta and subsequent transmission to the fetus [7]. Up to 40% of the survivors of prenatal or congenital infections suffer from varied sequelae including central nervous system abnormalities such as microcephaly and mental retardation; ocular abnormalities such as chorioretinitis and optic atrophy; sensorineural deafness, hepatosplenomegaly, pneumonitis, myocarditis, thrombocytopenic purpura, and hemolytic anemia [29,30,31]. Interestingly, perinatally infected children do not suffer from acute symptoms mentioned above, although rare cases of infantile pneumonitis have been reported [31].

## 3. Pathogenesis of CMV Retinitis

CMV is the largest Herpesviridae virus with double-stranded deoxyribonucleic acid (DNA) of 235 kbp [32]. The viral DNA is contained in an icosahedral protein capsid with a surrounding lipid bilayer [32]. The protein capsid and lipid bilayer facilitate invasion into the host with subsequent virion assembly and replication [32].

It is known that CMV establishes latency in blood and bone marrow of infected individuals, although the virus may not be isolated from host blood [33]. Viral reactivation occurs when the host immune system is severely compromised [1]. Studies have revealed that leukocyte depletion during blood transfusion drastically reduced CMV transmission, leading to further analysis that peripheral blood monocytes carried the latent CMV genome [34,35]. However, it was unclear how and when the latency was broken to cause recurrent CMV infections from the latent site [34,35]. Recent studies have now established that CD34+ progenitor cells form the basis for CMV latency and remain latent while they transform to undifferentiated monocytes [36,37]. Furthermore, CD34+ differentiation into dendritic cells is a prerequisite for CMV reactivation and release of infectious virus from their latent site [38,39]. In addition to blood, latent CMV DNA has been reportedly found in the lungs, brain, salivary glands, and the aqueous humor of the eye, creating a local latent site from which the virus may reactivate, symptomatically or asymptomatically [31,40,41,42]. Like many viruses that evade host cell immune defenses to enable survival, CMV has an interleukin 10 (IL-10) homolog that helps to prevent recruitment of natural killer and inflammatory cells [43]. While activated CMV is typically controlled by robust virus-specific CD4+ and CD8+ T-cell responses, under various forms of immunosuppression, including HIV/AIDS, CMV replicates and spreads rapidly to nearby tissue causing a variety of diseases [44,45].

Active CMV in the eye primarily infects vascular endothelial cells followed by retinal pigment epithelial cells in the retina causing viral cytopathic effect and subsequent retinal necrosis, as characterized in CMVR [25]. The large viral genome of CMV quickly translocates to the infected cell nucleus and starts production of viral progeny in a lytic lifecycle [24,25,32]. Viral progeny then bud out of the infected cell to infect neighboring cells and repeat the process. Uncontrolled replication of the virus leads to cell death in the involved retinal tissue leading to blurred vision, retinal detachment and ultimately blindness [25]. In a mouse model of CMVR; Chien et al., found that retinitis progression involves multiple pathways of cellular destruction involved in apoptosis, necroptosis, and pyroptosis and viral infection alone did not cause the cellular death and the clinical findings observed in CMVR [18].

## 4. CMVR in HIV

HIV causes progressive immune system failure once contracted. It is characterized by a single strand enveloped ribonucleic acid virus that, via reverse transcriptase, is converted to double stranded DNA and ultimately incorporates itself into the host genome, propagates and leads to T-cell depletion [46,47]. AIDS in HIV patients is defined by a CD4+ cell count <200 cells/mm^3^ or by the presence of an AIDS-defining illness, regardless of CD4 cell levels [20,46,47]. AIDS indicates advanced immunosuppression and allows for life-threatening complications including infections and malignancies to develop [20,46,47].

CMVR is the most common ocular opportunistic infection and the leading cause of blindness associated with AIDS worldwide [48,49,50,51,52]. CMVR occurs in approximately 20–40% of HIV-infected patients and accounts for 90% of HIV-related blindness [6,20,52]. In the literature to date, the most significant predictor for the development of CMVR is a CD4+ count less than 50 cells/µL [34,46,53,54,55]. While CMVR at higher CD4 counts is uncommon, a lower CD4+ T-lymphocyte count is strongly associated with increased retinal whitening and has been observed to be independent of other documented risk factors for CMVR [55,56]. CMV and HIV also have been found to transactivate each other leading to a greater degree of immune suppression in HIV patients, a possible explanation for why we see CMVR more commonly in HIV patients as opposed to non-HIV immune suppressed individuals [57]. Additional risk factors include lower CD8 T-lymphocyte counts, higher HIV blood level, lack of antiretroviral therapy, and male gender [55,56].

In addition to a low CD4+ cell count, individuals with low CD8+ cell counts are also at high risk for CMVR development and independently, CD8+ cells are predictive for CMVR development [55]. Specifically, patients with CD4+ cell counts <50 cells/µL and CD8+ cell counts <520 cells/µL may be at elevated risk [1,48,55]. HIV viral load levels may also impact CMV reactivation. It has been observed that HIV patients on combination antiretroviral therapy (ART) with a lower than 10-fold decrease in HIV viral loads during the treatment period are at elevated risk for opportunistic infections, including reactivation of CMVR [58]. Viral load is less predictive compared to CD4+ counts with mixed results in the literature; as in a prospective study of 102 patients, Song et al. did not demonstrate that HIV viral load predicted CMVR reactivation [2].

HIV-related retinopathy is an ocular risk factor for CMVR development. HIV retinopathy manifestations include cotton-wool spots (CWS), intraretinal hemorrhages, microaneurysms and exudates. Like CMVR, HIV retinopathy is also associated with lower CD4+ cell count and higher plasma HIV-RNA levels but is a non-infectious microangiopathy [3]. CWS in particular are important predictors of CMVR as a CWS represents an area of retinal microinfarction due to compromised circulation and may allow CMV to enter the retina and proliferate [50]. Due to this association with HIV retinopathy and subsequent CMVR, routine monitoring of HIV retinopathy may allow for early diagnosis of CMVR [3].

## 5. Non-HIV Pathologies Associated with CMVR

Non-HIV-related CMVR is less common than HIV-associated CMVR. CMVR in non-HIV cases have been observed in patients undergoing immunosuppressive therapies for hematologic malignancies, autoimmune disease, bone marrow and solid organ transplantation, and after local steroid administration [59]. The precise rates have not been well established due to its rarity. This is likely multifactorial and due to the higher immune function as compared to HIV patients where CMVR is strongly linked to CD4 cell count depletion.

### 5.1. Solid Organ and Bone Marrow Transplant

The most common non-HIV form of immunosuppression associated with CMVR occurs in the setting of organ transplantation [11,12,23,40,60,61,62]. CMVR following solid organ transplantation is rare but has been documented in kidney, heart, and liver recipients. CMV is the most common infection following kidney transplant and may cause glomerulonephritis, with reports of associated CMVR [61,62]. CMVR has been also been reported in rare cases following cardiac transplantation [60]. Additional risk factors for CMVR noted in cardiac transplant patients include microvascular disease secondary to hypertension, diabetes, and smoking. CMVR observed in these patients is likely primary CMV infection from the allograft or reactivation in the recipient host following subsequent medical immunosuppression [63]. CMVR has also been observed in bone marrow transplant (BMT) and hematopoietic stem cell transplantation (HSCT) recipients [63,64,65,66]. Similar to solid organ transplant patients, recent progress in HSCT and BMT matching and treatment regimens has resulted in prolonged survival in an immune suppressed state leading to a higher incidence of CMVR [63,64,65,66].

### 5.2. Hematologic Malignancy

CMVR has been associated with hematologic malignancies due to the immunocompromised status caused by the pathogenesis of the disease itself. In a report of three patients with chronic lymphocytic leukemia who developed CMVR, two patients had decreased CD4 counts [67]. Leukemic and lymphomatous infiltration of the eye may be difficult to distinguish from CMVR due to the overlap in retinal features observed in these patients. Analysis of intraocular fluid may be required for diagnosis in these patients [67,68]. Reports of CMVR in pediatric patients with acute lymphocytic leukemia have also occurred while patients are on maintenance immune suppressive therapy. There may be a delay in T-cell regeneration leading to CMV infection [69]. Low hemoglobin concentration has been found to be a statistically significant predictor of CMVR [70]. The exact reason for this association is unclear but a decreased life span of erythrocytes has been observed with infectious and immunologic disorders [70]. Due to the innate impaired hematologic system in these patients, they have inherent risk for CMVR occurrence [70].

### 5.3. Medication-Induced Immunosuppression

CMVR following periocular or intraocular steroid injection has been documented in immunocompetent individuals. CMVR following sub-Tenon triamcinolone injection, intravitreal dexamethasone implant, and fluocinolone acetonide implant in immunocompetent individuals on no other immunosuppression have been reported [71,72,73,74,75,76]. Local immunosuppression following intraocular or periocular steroid may result in reactivation of latent CMV resulting in CMVR [75,76].

Case reports of CMVR developing following systemic immunosuppression with corticosteroids and steroid-sparing agents have also been reported [77,78,79,80]. A case of CMVR in an HIV negative patient was reported following long-term systemic prednisolone for severe asthma; this patient had decreased T cell response to antigens [79]. CMVR has also been reported in patients using oral corticosteroids and azathioprine or mycophenolate mofetil for systemic lupus erythematosus in spite of normal CD4 T cell counts [81]. It is theorized these patients had a reduction in T-cell function resulting in CMVR. CMVR has also been observed with selective IgG2 deficiency following corticosteroids for inflammatory pulmonary disease [82].

## 6. Genetic Risk Factors

It is now known that genetic factors play a role in the susceptibility for CMV infection, CMVR progression and HIV mortality. Host genetic factors such as specific haplotypes of interleukin 10 receptor (IL-10R1), chemokine receptor 5 (CCR5), and stromal derived factor 1 (SDF-1) have been evaluated [83,84].

The CMV genome contains a human IL10R1 homologue that CMV uses to evade the human immune system. Sezgin et al. found that an amino acid changing variation in the cytoplasmic domain of IL-10R1 is protective against CMVR; however, the haplotype carrying an amino acid changing variation in the extracellular domain increases susceptibility to CMVR. There were variations in ethnicity, and this was observed in European-Americans but not African-American patients [83].

CCR5 is a receptor involved in HIV cellular invasion. In another study, Sezgin et al. again found variations based on ethnicity. It was observed that European-American CMVR patients with the CCR5+.P1+ promoter haplotype had increased mortality compared to those without this haplotype; however, this haplotype did not have a significant effect on CMVR progression or retinal detachment occurrence in this population. African-American patients with the CCR5+.P1+ promoter haplotype had increased risk for CMVR progression and demonstrated a trend towards greater mortality risk [84]. Increased risk of CMVR progression was also observed with the SDF1-3′A variant in the African-American cohort. SDF-1, serves as a ligand for the chemokine receptor (CXCR4), a co-receptor for HIV strains. Polymorphisms of SDF-1 also influence HIV course, AIDS progression, ART response and increased risk of retinal detachment [84]. Polymorphisms, such as the interferon lambda 3/4 variant have also been reported to increase susceptibility to CMV replication in organ transplant patients not on antiviral therapy and CMVR in HIV patients [85].

Although a number of candidate gene studies have been performed to determine other genetic polymorphisms that may increase susceptibility to CMV infection, most of the studies were limited by small sample size and have not yet been supported by genome wide association studies. 

## 7. CMV Retinitis in Children

CMV is the most common intrauterine viral infection affecting between 0.5–2% of all live births [86]. While CMVR affects 20%–40% of adults with HIV, it has only been reported in 5% of HIV-positive children [87]. In adults, CMVR presents in the peripheral retina in 85% of cases with isolated macular involvement occurring in less than 5%. Conversely, infants with CMVR have a predilection for the disease to present in the macula. In children, bilateral CMVR is more common, occurring in up to 89% of pediatric cases compared to only 33.5% of adult cases [51,87]. CMVR in children tends to occur in patients with CD4 count below 20 cells/µL. Thus, routine ophthalmologic screening exams may not detect CMVR until CD4 count is below 20 cells/µL or a CD4/Absolute lymphocyte count is less than 2%. Since children with even advanced CMVR may not complain of vision loss, frequent ophthalmologic screening should be done once these thresholds are reached to allow for earlier detection and initiation of treatment [49,88].

The differences observed in CMVR in children compared to adults may be related to the fact that children have primary CMV infection rather than reactivation of latent infection. Children′s immune systems are also immature therefore the systemic response to CMV may be more severe [87].

## 8. Clinical Presentation: HIV and Non-HIV Patients

In more than half of infected patients, CMVR is asymptomatic and is revealed during routine ophthalmic examination [49,89]. When present, ocular symptoms include decreased visual acuity, flashing lights, scotomas and floaters. Of these symptoms, blurred vision is the most common ocular symptom experienced by patients with CMVR [89,90]. As early cases tend to be asymptomatic, symptoms tend to translate into more severe retinitis, early involvement of the macula, or possible infiltration of the optic nerve [48,49]. Kim et al. found that the vision at presentation did not significantly differ between the HIV and non-HIV immunosuppressed patients [6]. Lu et al. did note worse presenting vision in non-HIV immunosuppressed patients, most likely because their patients were significantly immunocompromised, as the majority of them had hematologic malignancies and had received intensive chemotherapy shortly before presentation [4].

There are three main clinical variants of CMVR [91]. The first is an indolent variant where the main retinal manifestations are granular, white areas of progressive necrotizing retinitis with minimal to no hemorrhage, often in the peripheral retina (Figure 1). The second is a fulminant form characterized by full-thickness yellow-white lesions with retinal hemorrhage in a sectoral, perivascular distribution (Figure 2). The third variant is a perivascular form, also described as “frosted branch angiitis”. The most distinctive feature of CMV retinitis is an opaque, white lesion border, reflecting retinal edema and necrosis. This opacification tends to be greater with patients with greater degrees of immunosuppression [21]. As CMVR progresses, the retina becomes atrophic and can develop tears which can result in retinal detachment. 

CMVR is more commonly seen in HIV patients, likely due to the degree of immune suppression required for manifestation and there are few studies that directly compare HIV and non-HIV CMVR patients [6,91]. When classic retinitis findings are present, the prevalence of fulminant versus indolent forms of retinitis have been observed to be similar in HIV and non-HIV patients [91]. Kuo et al. evaluated CMVR in non-HIV immunosuppressed patients and found that retinal features were similar to retinal features observed in HIV CMVR patients in the literature [59]. These similarities observed in CMVR in non-HIV to those patients with HIV are theorized to be due to similar mechanisms resulting from decreased T cells [91,92,93]. Lu et al. noted, however, that non-HIV immunocompromised-related retinitis had a more aggressive presentation with more involvement of the posterior retina and had greater retinal involvement, a finding that has not been replicated. In contrast, Kim et al. in 78 eyes from 58 patients found that bilateral involvement and posterior involvement did not differ between the HIV and non-HIV groups [6]. These mixed results in terms of degree and location of retinitis may be attributed to the level of immunosuppression [94]. 

It has also been observed that vitreous inflammation, arteritis, and vascular occlusion outside of the areas of active retinitis can be more severe in non-HIV patients and the degree may vary depending on the level of immune suppression in non-HIV immunosuppressed patients [21,54,91,93]. Non-HIV immunosuppressed patients have been found to demonstrate higher levels of CD4 immune response possibly explaining the lower incidence of CMVR in this group as compared to HIV-immune suppressed individuals [95]. Immunological profiles have been evaluated in non-HIV and HIV patients (including CD4+ and CD8+ T lymphocyte counts and CD4+/CD8+ cell ratios) and average cell counts in non-HIV patients with CMVR tend to be within normal limits [95]. Similarly, Bronke et al. found that HIV patients lose CMV-specific CD4 + cells allowing for greater CMV dissemination compared to in those with immune suppression from other causes [96]. 

While CMV predominantly affects the posterior segment, additional clinical findings in immune-deficient patients with CMVR include diffuse, stellate corneal endothelial deposits that can be detected on slit-lamp examination. In a post-mortem evaluation in these patients, it was found that these deposits are composed of macrophages and fibrin and are devoid of lymphocytes [97]. The prevalence of macrophages observed on histopathology may be related to the inability of the immunodeficient patient to mount a normal T-cell response as compared to immune competent individuals. Immune competent individuals can develop CMV anterior uveitis with similar anterior segment signs of inflammation with stellate keratic precipitates and anterior chamber inflammation without associated retinal findings [97].

Following institution of antiretroviral therapy, increased immunocompetency may cause an inflammatory response to CMV antigens. Immune reconstitution inflammatory syndrome (IRIS) is a phenomenon where increased inflammation is observed in HIV patients who are on ART during the initial weeks of treatment. This exaggerated reactivation of the immune system can lead to worsening reaction to CMV [19,51,98]. Immune recovery uveitis (IRU) is the predominant form of ocular immune reconstitution syndrome [98,99,100]. IRU can result in visual loss from macula edema, retinal neovascularization and cataracts, in the absence of active CMV disease [99,100]. IRU is common with twelve (89.7%) of 14 patients developing uveitis prior to discontinuation of anti-CMV therapy in one study [101]. Risk factors for IRU include immune reconstitution with ART, more extensive CMV lesions and the use of cidofovir [99].

## 9. Diagnostic Tools

The diagnosis of CMVR is primarily clinical and based on the observation of characteristic retinitis on dilated funduscopic examination by a trained ophthalmologist. Polymerase chain reaction (PCR) analysis of aqueous or vitreous ocular fluid can further confirm the diagnosis and may be helpful in cases with atypical clinical presentations. It can help differentiate CMVR from retinitis caused by other members of the Herpesvirus family, such as varicella zoster or herpes simplex virus, or from other pathogens such as *Toxoplasma gondii* [102,103].

Isolation of CMV from blood leukocytes can provide insight into active shedding of the virus; however, this test is not sensitive and may require multiple weeks to become positive. Patients who are immunosuppressed may not develop detectable titer levels [102,103]. Serum CMV antigenemia and PCR tests are sensitive measures that may predict CMV disease up to several months prior to clinical manifestation [102,103]. An antigen level less than 45 strongly suggests the absence of retinitis. The negative and positive predictive values of the CMV antigen test were 98.2% and 80%, respectively [104].

Another study suggested ophthalmic screening in HIV patients with CMV present in urinalysis or “CMVuria”; CMVuria as a single finding, however, does not justify antiviral prophylaxis against CMVR [105]. CMV antigenemia testing may be a valuable tool for the rapid diagnosis of CMV disease in HIV-infected individuals, but these results need to be interpreted in the context of the clinical presentation and ophthalmic findings [103,106].

## 10. Management of CMVR

In both HIV and non-HIV patients, high dose induction antiviral therapy is initiated when active CMVR is diagnosed. Induction therapy is typically administered for 14 to 21 days but the final duration is determined by the clinical response to therapy. Induction is followed by continuous maintenance therapy until CD4 count increase in HIV patients is observed, when ART is therapeutic, and/or when CMVR shows no progression [107]. In HIV patients, management hinges on ART optimization as immune restoration alone can result in resolution [57,107,108]. 

CMVR management involves intravenous (IV), oral, and intravitreal injections (IVI) of antiviral medications. The location of the CMVR lesions largely dictates the treatment algorithm. For patients with immediate sight-threatening lesions, intravitreal injections in conjunction with systemic therapy are currently recommended. For patients without immediately sight-threatening lesions, systemic therapy alone with close observation is reasonable. The main virostatic drugs employed today include valganciclovir (oral), ganciclovir (IV, IVI), foscarnet (IV, IVI), and Cidofovir (IV, IVI). Acyclovir is not used in the treatment of CMV as this drug specifically requires phosphorylation by viruses to become active, a mechanism which CMV is not capable of as it does not encode for virus-specific thymidine kinase [109]. Comparison studies of various systemic anti-CMV agents have not shown superiority of one agent over another. The choice of which antiviral agent to use is multifactorial and is influenced by the ability to tolerate oral medications, comorbid conditions and medications, and predicted or observed compliance with therapy [57,110,111]. 

Ganciclovir was the first antiviral agent approved for the treatment of CMV [1]. The primary mechanism of ganciclovir action is inhibition of the replication of CMV DNA via DNA polymerase by ganciclovir-5′-triphosphate [110]. Ganciclovir is given IV rather than orally due to poor bioavailability with oral administration [57]. Valganciclovir is an oral prodrug that is converted to ganciclovir in the body [111]. Oral valganciclovir is an efficacious treatment option in both HIV and non-HIV CMVR patients. Oral valganciclovir was approved for use for CMVR in 2000, can be used for induction and maintenance therapy, and has an excellent absorption profile and subsequent systemic drug concentrations [1]. Oral valganciclovir avoids complications associated with intravenous formulations that require in-dwelling catheters [57]. In a multicenter randomized trial performed in 2002 evaluating 160 patients with AIDS and newly diagnosed CMV retinitis, induction therapy with valganciclovir was found to be equally efficacious as IV ganciclovir [111]. This study excluded patients with centrally located CMVR; therefore, adjunct IVI is still utilized in immediate sight-threatening disease.

IV ganciclovir or foscarnet are effective options in individuals who are unable to tolerate oral therapy [112]. Foscarnet inhibits CMV DNA replication and reverse transcription of HIV [94,112,113,114]. Foscarnet has been effective in treating AIDS patients with rapidly progressing CMVR whose CMV isolates were resistant to ganciclovir in vitro. Results from a large multicenter clinical trial revealed that patients with AIDS treated with systemic foscarnet for CMVR had longer life expectancy compared to those who initially received ganciclovir [113]. Foscarnet is highly nephrotoxic, can cause electrolyte abnormalities, and may cause nausea and vomiting [114].

Cidofovir is a monophosphate nucleotide analogue. In the body, cidofovir becomes phosphorylated by intracellular kinases and competitively inhibits the addition of deoxycytidine triphosphate into viral DNA via DNA polymerase [115]. Due to systemic side-effects of nephrotoxicity requiring intravenous pre-hydration with normal saline and administration of probenecid with each infusion, cidofovir is currently not used as first line therapy [115].

IVI of virostatic medications can be considered for short-term management or as adjunctive therapy but sole intravitreal therapy is not commonly recommended except in circumstances where systemic therapy is contraindicated [110]. Local monotherapy may be utilized more frequently in post-transplant patients receiving immunosuppression as myelosuppression may worsen with systemic antiviral therapies. In non-HIV immune compromised patients, studies have shown IVI of ganciclovir once weekly for approximately 6 weeks showed successful resolution of CMVR [116]. However, in HIV patients, Jabs et al. found that compared with systemic treatment, regimens containing only intravitreal injections had greater rates of retinitis progression [57]. Complications associated with IVI include increased intraocular pressure, retinal detachment, vitreous hemorrhage, and intraocular infection. Ocular side effects of IVI and IV cidofovir specifically include anterior uveitis and hypotony [110]. Cidofovir-associated uveitis seems to occur more frequently in AIDS patients with CMVR who are at risk of being unable to restore immunity on ART [117]. 

In recalcitrant cases or relapsed infection, combination therapy is recommended. It has been observed that patients treated with both ganciclovir and foscarnet had a longer life expectancy than those given either drug alone. No additional toxicity was observed with combination of foscarnet and ganciclovir in patients with severe CMV disease [94,113].

Letermovir is an antiviral drug that, as of 2017, has been approved for prophylaxis in CMV-seropositive allogeneic hematopoietic-cell transplantation recipients. Letermovir inhibits the CMV–terminase complex and in the phase 3 randomized, double-blind, placebo-controlled, superiority trial, letermovir prophylaxis resulted in a significantly lower risk of clinically significant CMV infection compared to placebo [118]. Valganciclovir and ganciclovir are routinely used in solid-organ transplantation CMV prophylaxis but use may be limited by myelosuppression after hematopoietic-cell transplantation. Ongoing studies are being conducted for Letermovir in solid-organ transplant patients [119].

There are a number of patents for new drugs evaluating gene regulation and molecular biology. Medications targeting ribonucleic acid (RNA) and blocking translation of RNA into proteins required for replication of CMV are being evaluated as are other therapies targeting various aspects necessary for CMV survival and replication [1].

## 11. ART and CMVR

The epidemiologic features, incidence and prevalence of CMVR in HIV patients have changed greatly with the advent of anti-retroviral therapy [49,50,51,56,108]. ART is a combination drug regimen that classically includes two nucleoside analogues and a protease inhibitor but may also include a combination of nucleoside reverse transcriptase inhibitors, non-nucleoside reverse transcriptase inhibitors, protease inhibitors, integrase inhibitors, or fusion inhibitors that impairs HIV replication [51,99]. Successful ART treatment reduces the plasma HIV load and increases the CD4+ T-lymphocyte levels [57,58]. CMV and HIV transactivate one other giving an additive immunosuppressive effect [57]. In the pre-ART era, CMVR was characterized by relapsing retinal disease resulting in progressive visual loss and was associated with increased mortality [108]. This occurred despite the use of repeat induction anti-CMV therapy and maintenance treatment between CMVR episodes [108]. Following ART development, the incidence of CMVR has declined by approximately 75–90% [51,99]. With ART, a significant decrease in the incidence and recurrence of CMVR parallels an increase in CD4-lympoctye count and decrease in levels of plasma CMV-DNA, which tends to become undetectable with CD4+ counts above 100 cells/µL [51,54,108]. Patients with CMVR treated with ART also now have increased time to relapse and prolonged survival [51,58]. No prior antiretroviral regimen has produced such marked, sustained decrease in viral load suggesting that ART regenerates anti-CMV immunity [51,58].

## 12. Drug Resistance

Resistance to antiviral therapy for CMV can occur in patients receiving prolonged therapy with ganciclovir and valganciclovir. The duration of medication use, ongoing viral replication and degree of host defense influence drug resistance [120]. In AIDS patients, severely decreased CD4 cell counts and long-term anti-CMV therapy resulted in upwards of 20% incidence of resistance to all anti-CMV medications by 9 to 12 months in one study [121].

CMV virological factors influence drug resistance development. CMV latency with later reactivation allows for prolonged replication that can yield many CMV variants resulting in resistant strains [120]. Drug resistance generally results from mutations in two CMV genes, the phosphotransferase gene (pUL97) and the viral DNA polymerase gene (pUL54). Ganciclovir and valganciclovir require phosphorylation to function. Mutations in the UL97-encoded CMV phosphotransferase impairs phosphorylation therefore causing resistance. UL97 mutations first cause resistance to ganciclovir but not to foscarnet or cidofovir. With continued ganciclovir use, pUL54 may also develop resulting in significant resistance to ganciclovir, cidofovir, and foscarnet [120].

Patients with a detectable CMV viral load in the blood or ocular fluid can undergo genotype testing for drug resistant CMV via PCR. Real-time PCR assays can identify drug resistance mutations in UL97 by analyzing changes in melting curves, following amplification [120]. The advantages of this technique include that a small amount of viral DNA can be amplified without cell culture and mixed virus populations and multiple mutations can be detected simultaneously. This technique, however, cannot distinguish point mutations that occur in the same codon and polymorphisms near known mutations may affect the melting curve and requires individual probes [120]. An alternative test is a phenotypic assay known as the plaque reduction assay, where the ability of CMV to grow in the presence of an anti-CMV drug is the surrogate for resistance. The number of plaques that grow on varying drug concentrations is compared to the number of plaques produced in the absence of the drug. The concentration where there is a 50% reduction in plaque count is then calculated as the 50% inhibitory concentration for the drug in question. Phenotypic assays lack standardization, however, and take upwards of four weeks to perform [120].

New advancements in treating ganciclovir-resistant CMV are developing, including leflunomide and T-lymphocyte infusion. Leflunomide, an immunosuppressive therapy for rheumatoid arthritis, has been used as an off-label treatment for drug resistant CMV [122,123,124,125]. Leflunomide inhibits the viral nucleocapsid and tegument development, resulting is decreased cross-resistance from DNA polymerase antivirals. Successful treatment of CMV disease with leflunomide in solid organ transplant, stem cell transplant and multidrug resistant CMV have been observed [122,123,124,125].

T-lymphocyte infusion has also been successfully used in resistant cases. In a case report of CMV viremia in a patient with acute lymphoblastic leukemia, status post allogeneic stem cell transplant and on immunosuppressive therapy for graft-versus-host disease, the patient was originally treated with ganciclovir. Due to a confirmed CMV UL54 mutation resistant to foscarnet, cidofovir, and ganciclovir, the patient was switched to leflunomide [126]. The patient then developed CMVR in his fellow eye. Antiviral therapy was discontinued, and he was treated with partially HLA-matched CMV pp65-specific cytotoxic T-cells from a donor. Following six infusions, there was complete resolution of both the retinitis and viremia with an undetectable CMV viral load through the three month follow up period. However, given the complexity of these treatments, it is currently only recommended for those with resistance to intravitreal and systemic therapies [126].

## 13. Discontinuation of Therapy

Prior to the development of ART, CMVR in an HIV patient required life-long anti-CMV therapy [127]. Many recent studies suggest that CMVR maintenance therapy discontinuation is safe for patients with quantitative immune recovery in CD4 counts with ART treatment, even with a history of severe immunosuppression [51,127,128]. Current guidelines for discontinuing maintenance therapy include a sustained rise in CD4+ cells >100 cells/µL, CD4+ rise must be at least 50 cells/µL, and relapse-free intervals of 3–6 months must be observed with inactive CMVR characterized by retinal scarring [51,127,128,129]. It is important to closely monitor these patients with dilated ophthalmic examinations as ART failure may occur and allow CMVR reactivation (Figure 3) [127]. The revised 2015 CDC guidelines for CMVR in HIV patients describe that routine screening in patients with CD4+ counts <50 cells/µL every 3–4 months was advised in the pre-ART era but no definitive post-ART examination schedule has been established. These guidelines also outline that valganciclovir primary prophylaxis against CMVR is not required once CMVR treatment is successfully completed unless CD4 count has decreased to <100 cells/µL [129].

In non-HIV and HIV patients on anti-CMV drug treatment, resolution of the retinal opacity with resultant atrophy is considered a reliable sign of CMV inactivity [4,53]. One study followed 20 non-HIV immunosuppressed patients for 17 months and found a recurrence rate of 33.3% occurred following discontinuation of anti-CMV therapy; as such, routine ophthalmic monitoring is advisable [4]. Similarly, Kuo et al. found that after immunosuppressive therapy was discontinued in non-HIV immunosuppressed patients, 56% were able to discontinue anti-CMV therapy and had no subsequent CMVR reactivation [59]. 

## 14. Prognosis

CMVR may cause debilitating, typically permanent vision loss, which is largely preventable by routine monitoring in immunosuppressed patients and ART initiation in HIV patients. Independent risk factors may aid in predicting the final visual outcome of CMVR and include extent of retina infected with CMV, better presenting visual acuity and absence of macular involvement. Poor prognostic factors include the increased size and activity of lesions, lack of ART in HIV patients during infection onset and lack of ART-associated immune reconstitution [6,130].

The overall visual prognosis and clinical features of CMVR have not been found to differ between HIV and non-HIV patients [4,6]. Despite advances in anti-viral treatment, the visual prognosis of CMVR remains poor [6,92]. Poor prognosis is associated with retinal detachment during treatment, macular involvement and poor general health of the patient [6]. The incidence of retinal detachment also has not been found to significantly differ comparing HIV and non-HIV CMVR patients except in one small case series where HIV patients had higher rates of retinal detachment and more clock hours of retinitis on presentation than non-HIV CMVR patients [131]. Kuo et al. reported that the overall clinical course of CMVR in non-HIV patients is similar to CMV retinitis in HIV-patients. ART therapy has led to a decrease in the incidence of CMVR in HIV patients; however, upwards of 20% of HIV patients still become legally blind in one or both eyes due to CMVR [59].

## 15. Conclusions

CMV retinitis is one of the most serious ocular complications in immune-suppressed individuals and can lead to irreversible vision loss. CMVR can be prevented with ART optimization early in the course of HIV disease; however, with the increased life expectancy of HIV patients, CMVR is still a problem today. With every increasing indication for immune suppression, CMVR is also presenting more frequently in non-HIV immune suppressed individuals. It is exceedingly rare to observe CMVR in healthy patients unless it is secondary to local ocular steroid treatment [71,72,73,74,75,76].

There are many overlapping characteristics of CMVR in HIV patients and non-HIV immune suppressed patients. The retinal manifestations are similar; however, non-HIV immune suppressed patients tend to have a more robust inflammatory response in the other segments of the eye [6]. Treatment in these two cohorts both hinge on induction and maintenance therapy with antiviral medications with differences in dosing strategies and indications for treatment discontinuation [51,127,128,129,130]. Improved anti-viral treatment modalities will continue to increase the survival of HIV and non-HIV patients with CMVR alike. Progression of the retinitis frequently still occurs while on current therapeutic regimens in many patients and as survival increases, ocular complications may be seen more frequently over time [132,133]. Experts today generally recommend asymptomatic screening in those at high risk (CD4+ cells <100 cells/µL or CMV viremia on PCR) with dilated fundus examinations in order to diagnose the disease at an early stage [129]. Screening programs in non-HIV immune suppressed patients are more variable and further identification of risk factors may help develop screening guidelines. 

## Figures and Tables

**Figure 1 microorganisms-08-00055-f001:**
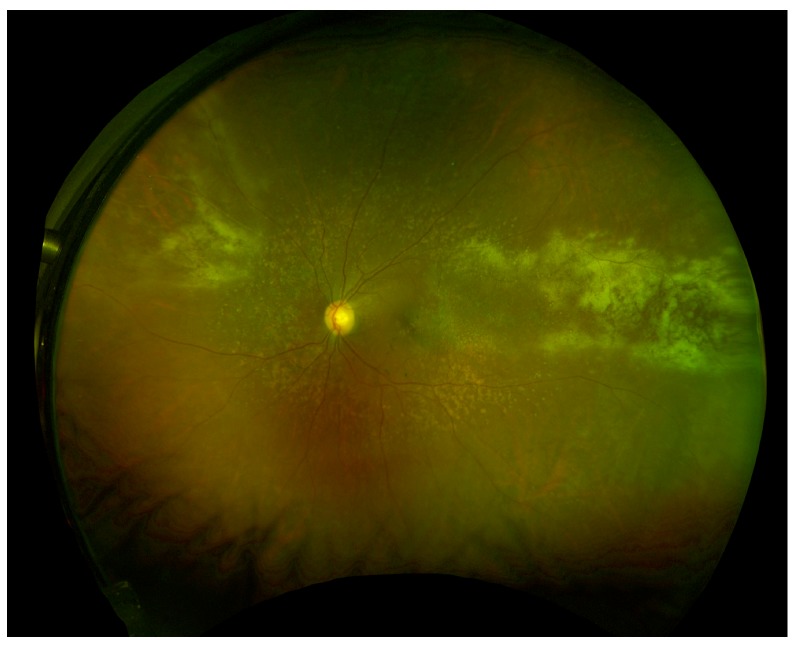
Wide-field fundus image of a patient status post renal transplant on immunosuppression with indolent form of CMVR. (*Image courtesy of Dr. Felix Chau and Dr. Pooja Bhat).*

**Figure 2 microorganisms-08-00055-f002:**
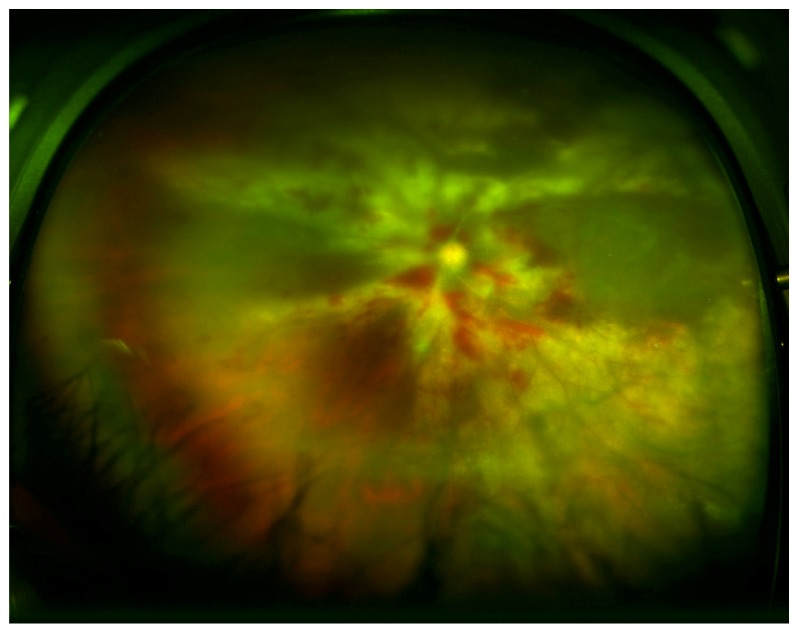
Wide-field fundus image of a patient with lymphoma with fulminant form of CMVR with widespread retinal whitening, necrosis, and hemorrhage. (*Image courtesy of Dr. William Mieler).*

**Figure 3 microorganisms-08-00055-f003:**
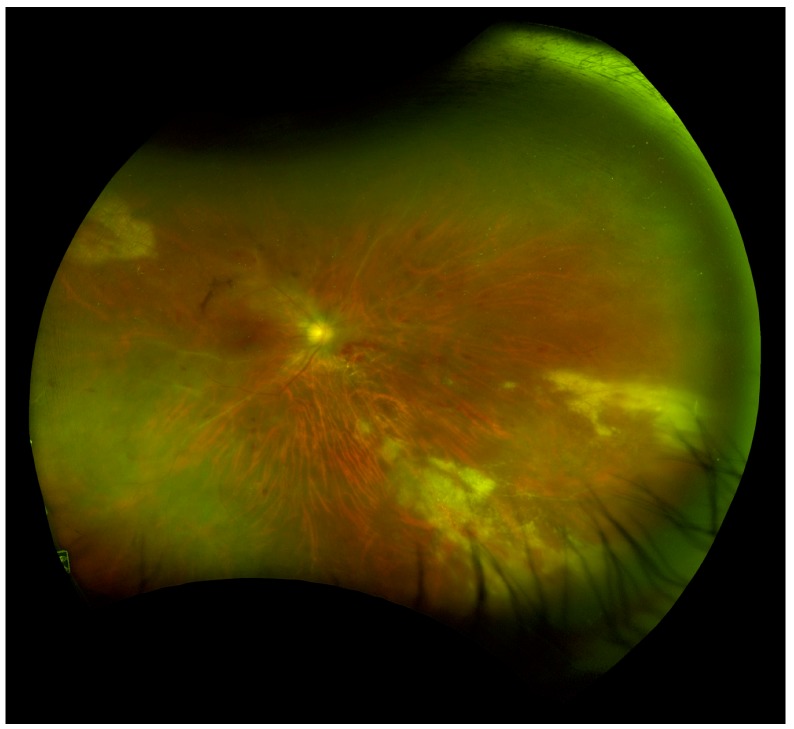
Wide-field fundus image of a patient with HIV with CMVR recurrence and immune recovery uveitis (IRU) requiring intravitreal and systemic anti-viral therapy. *(Image courtesy of Dr. Pooja Bhat).*

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
