# Peer review of "Cytomegalovirus Retinitis in HIV and Non-HIV Individuals"

_microorganisms, 2019, doi:10.3390/microorganisms8010055_

Round 1

Reviewer 1 Report

“Cytomegalovirus retinitis in HIV and non-HIV individuals” is a well written review by Munro and colleagues that provides the reader with a comprehensive overview of this sight-threatening retinal disease in the era of antiretroviral therapy. Overall, it is nicely divided into thoughtful sections that are focused on various aspects of cytomegalovirus infection and disease. Nonetheless, some inaccuracies and incomplete discussions present themselves.

While CMV retinitis develops in HIV-immunosuppressed persons with AIDS as well as in persons immunosuppressed for solid organ or bone marrow transplantation, the percentage of CMV retinitis seen in these two patient populations differ greatly (i.e., ~20 to 40% vs ~5%, respectively). This clinical observation is not addressed nor is its relevance to the pathogenesis of CMV retinitis in persons suffering from different patterns of immune suppression.

Paragraph 2 of the Introduction section is the “Materials and Methods” section of the review, but is distracting, acting as a “speed bump” for the reader. Perhaps its location would best be served by placing it following the main text of the paper (after the Conclusions section) where the Materials and Methods section for publications in many journals is now found. It could be placed as a paragraph with Author Contributions, Funding, Acknowledgments, and Conflicts of Interest.

What percentage of the world population is seropositive? This important information is apparently overlooked. See the recent review by Zuhair et al in Rev Med Virol 2019; e2034.

Line 49 should include the fact that CMV also remains latent in bone marrow cells.

The first paragraph of Section 2 is a mass of information that could be better communicated using separate paragraphs focused on individual topics. Specifically, there is a need for one paragraph devoted to congenital CMV disease. As written, this important information is buried in one paragraph on pathogenesis and lost to the reader.

The use of the term “necrosis” in Line 94 is outdated. Firstly, mounting evidence in the literature suggests that “necrosis” is actually the programmed cell death pathway “necroptosis”. Secondly, several pathogenic mechanisms including a number of cell death pathways conspire to cause the retinal tissue destruction observed during onset and progression of CMV retinitis.

The term “highly active retroviral therapy” or “HAART” is outdated and has been replaced by “combination antiretroviral therapy” or “ART” to reflect the many possible anti-HIV drugs available to the clinician. These now include nucleoside reverse transcriptase inhibitors, non-nucleoside reverse transcriptase inhibitors, protease inhibitors, intregrase inhibitors, and even fusion inhibitors.

The term “strains” is used on Line 279 to indicate other human herpesviruses such as VZV and HSV. This is inaccurate. VZV and HSV are other members of distinct Subfamilies found within the Herpesviridae Family. “Strains” refer to individual clinical isolates. The authors are referred to Fields Virology.

Because CMV does not encode for thymidine kinase, acyclovir is not used for management of CMV infections. This information should be provided in Section 10.

Author Response

Reviewer Comments and Writer Responses

We thank the reviewers for their comments and offer our point by point responses below:

Reviewer 1

While CMV retinitis develops in HIV-immunosuppressed persons with AIDS as well as in persons immunosuppressed for solid organ or bone marrow transplantation, the percentage of CMV retinitis seen in these two patient populations differ greatly (i.e., ~20 to 40% vs ~5%, respectively).

Thank you for your review and this comment. We have added additional information throughout sections 4 (line 159-162) , 5 (lines 183-189), and 8 (lines 321-328.

Paragraph 2 of the Introduction section is the “Materials and Methods” section of the review, but is distracting, acting as a “speed bump” for the reader. Perhaps its location would best be served by placing it following the main text of the paper (after the Conclusions section) where the Materials and Methods section for publications in many journals is now found. It could be placed as a paragraph with Author Contributions, Funding, Acknowledgments, and Conflicts of Interest.

Thank you. This paragraph has been moved to the end of the manuscript.

What percentage of the world population is seropositive? This important information is apparently overlooked. See the recent review by Zuhair et al in Rev Med Virol 2019; e2034.

This information has been added at line 43 under the pathogenesis section.

Line 49 should include the fact that CMV also remains latent in bone marrow cells.

This information has been added at line 89 and 118.

The first paragraph of Section 2 is a mass of information that could be better communicated using separate paragraphs focused on individual topics. Specifically, there is a need for one paragraph devoted to congenital CMV disease. As written, this important information is buried in one paragraph on pathogenesis and lost to the reader.

Thank you – we have re-worked this paragraph to improve the readability and have added additional information (see new sections 2A and 2B).

The use of the term “necrosis” in Line 94 is outdated. Firstly, mounting evidence in the literature suggests that “necrosis” is actually the programmed cell death pathway “necroptosis”. Secondly, several pathogenic mechanisms including a number of cell death pathways conspire to cause the retinal tissue destruction observed during onset and progression of CMV retinitis.

Thank you. We have updated the terminology and have added the citation: H. Chien, R.D. Dix Evidence for multiple cell death pathways during development of experimental cytomegalovirus retinitis in mice with retrovirus-induced immunosuppression: apoptosis, necroptosis, and pyroptosis, J. Virol., 86 (20) (2012), pp. 10961-10978

The term “highly active retroviral therapy” or “HAART” is outdated and has been replaced by “combination antiretroviral therapy” or “ART” to reflect the many possible anti-HIV drugs available to the clinician. These now include nucleoside reverse transcriptase inhibitors, non-nucleoside reverse transcriptase inhibitors, protease inhibitors, intregrase inhibitors, and even fusion inhibitors.

This has been corrected throughout the manuscript.

The term “strains” is used on Line 279 to indicate other human herpesviruses such as VZV and HSV. This is inaccurate. VZV and HSV are other members of distinct Subfamilies found within the Herpesviridae Family. “Strains” refer to individual clinical isolates. The authors are referred to Fields Virology.

This has been corrected (Line 355).

Because CMV does not encode for thymidine kinase, acyclovir is not used for management of CMV infections. This information should be provided in Section 10. 

This information has now been added in Section 10. Thank you.

Reviewer 2 Report

Munro et al, review the pathogenic and clinical characteristics of cytomegalovirus-induced retinitis in both HIV- and non-HIV individuals. Overall, the review is informative, well researched, constructed in a logical manner and provides valuable information regarding a disease whose profile has fallen with the advent of highly active retrovirus therapy.

The principal criticism concerns the “Pathogenesis” section which is somewhat brief and superficial since it does not discuss animal models of this disease. Murine cytomegalovirus infection of immunosuppressed mice has been used as a model system to investigate the ocular pathogenesis of cytomegalovirus infection and has provided some valuable insights into the course of retinal virus dissemination as well as mechanisms of retinal cell death. For instance, it has been shown that death of uninfected “bystander cells” is a major cause of CMV-induced retinal pathology. A discussion of these studies would strengthen the review.

There are only minor issues with grammar and spelling.

Author Response

We thank the reviewers for their comments and offer our point by point responses below:

REVIEWER 2

The principal criticism concerns the “Pathogenesis” section which is somewhat brief and superficial since it does not discuss animal models of this disease. Murine cytomegalovirus infection of immunosuppressed mice has been used as a model system to investigate the ocular pathogenesis of cytomegalovirus infection and has provided some valuable insights into the course of retinal virus dissemination as well as mechanisms of retinal cell death. For instance, it has been shown that death of uninfected “bystander cells” is a major cause of CMV-induced retinal pathology. A discussion of these studies would strengthen the review

Thank you – please see the discussion added at the end of the pathogenesis section 2 and 3 with added references on animal models of disease.

Round 2

Reviewer 1 Report

The authors have responded to all suggestions and questions in a thoughtful and thorough manner. The revised review is therefore greatly improved when compared with the original submission.

Reviewer 2 Report

No further question